# Distributed Fibre Optic Sensing (DFOS) for Deformation Assessment of Composite Collectors and Pipelines

**DOI:** 10.3390/s21175904

**Published:** 2021-09-02

**Authors:** Bartosz Bednarz, Paweł Popielski, Rafał Sieńko, Tomasz Howiacki, Łukasz Bednarski

**Affiliations:** 1Faculty of Building Services, Hydro and Environmental Engineering, Warsaw University of Technology, Nowowiejska St. 20, 00-653 Warsaw, Poland; pawel.popielski@pw.edu.pl; 2Reinforced Concrete Structures Division, Faculty of Civil Engineering, Cracow University of Technology, Warszawska St. 24, 31-155 Cracow, Poland; rafal.sienko@pk.edu.pl (R.S.); th@shmsystem.pl (T.H.); 3SHM System Sp. z o.o., Sp. kom., Libertów, ul. Jana Pawła II 82A, 30-444 Krakow, Poland; 4Department of Mechanics and Vibroacoustics, Faculty of Mechanical Engineering and Robotics, AGH University of Science and Technology in Krakow, Mickiewicza Al. 30, 30-059 Krakow, Poland; lukaszb@agh.edu.pl

**Keywords:** glass fibres, stress concentrations, mechanical testing, process monitoring, distributed fibre optic sensors, strains, displacements, shape sensing

## Abstract

Due to the low costs of distributed optical fibre sensors (DFOS) and the possibility of their direct integration within layered composite members, DFOS technology has considerable potential in structural health monitoring of linear underground infrastructures. Often, it is challenging to truly simulate the actual ground conditions at all construction stages. Thus, reliable measurements are required to adjust the model and verify theoretical calculations. The article presents a new approach to monitor displacements and strains in Glass Fiber Reinforced Polymer (GFRP) collectors and pipelines using DFOS. The research verifies the effectiveness of the proposed monitoring solution for health monitoring of composite pipelines. Optical fibres were installed over the circumference of a composite tubular pipe, both on the internal and external surfaces, while loaded externally. Analysis of strain profiles allowed for calculating the actual displacements (shape) of the pipe within its cross-section plane using the Trapezoidal method. The accuracy of proposed approach was positively verified both with reference spot displacement transducer as well as numerical simulations using finite element method (FEM). DFOS could obtain a comprehensive view of structural deformations, including both strains and displacements under externally applied load. The knowledge gained during research will be ultimately used for renovating existing collectors.

## 1. Introduction

An increase in the application of high-tech solutions, particularly aimed at optimising financial expenditures related not only to the process of construction itself, but also to the entire lifecycle of the structure [1], is being observed in the design, construction and operation of engineering structures in recent years. New composite materials with very good mechanical parameters and the use of prefabrication advantages will find wider and wider application in civil engineering and geotechnics [2,3,4]. Such materials include GFRP (Glass Fiber Reinforced Plastic) composites, which due to their low self-weight, high strength [5], resistance to corrosion, resistance to harsh environmental conditions [6,7] and, in particular, very high durability and long-term performance, find increasingly broad application in many areas of engineering [4,8].

The introduction of new technologies to the industry should always be accompanied by detailed laboratory research and preferably by verifying the behaviour of real structure operated in in situ conditions. Therefore, there is a strong need to generate new measurement solutions that provide as much comprehensive and reliable information as possible—along with simultaneous financial feasibility [5]. This criterion, particularly in reference to linear structures such as pipelines, is currently met by the distributed fibre optic sensing technology [9,10], as it allows for measurements of strains [11], displacements (shapes) [12,13], as well as temperatures [14] over the entire length of the optical fibre sensor [15]. Furthermore, such an approach enables effective and economic replacement of thousands of traditional spot strain gauges [16]. Such an approach offers completely new possibilities in analysing and assessing technical condition of the structure in question.

The article presents an example of laboratory tests where composite pipeline specimens made from GFRP were loaded within a testing machine. Such composite structural components are very often used to strengthen existing concrete or masonry collectors [17] by placing them inside the collectors and injecting the space between the old and new parts. Due to the safety-critical target application, the studies carried out are of significant practical importance. Thus, it was decided to analyse structural deformation using distributed fibre optic sensing (DFOS) with extremely high spatial resolution (starting from as fine as 5 mm). To do so, optical sensors were installed over the composite specimen circumferences, both in the internal and external surfaces. Detailed analysis of measured strain profiles allowed for calculating the real displacements (shape) of the pipe within its cross-section plane. Having strain and displacement values at each point of the circumference under known load, it was then possible to compare and calibrate results obtained from theoretical simulations based on numerical (FEM) model. Moreover, it was possible to verify the effectiveness of the proposed monitoring solution for structural control of the composite pipeline or collectors.

## 2. Structural Health Monitoring of the Pipelines

The operation of pipeline structures bears very high risk due to extreme consequences of their damage, including financial, environmental and social consequences. In addition, there are many random phenomena that can occur during several decades of pipeline operation over lengths of even hundreds of kilometers, and which cannot be predicted theoretically during the design stage. An example of such an effect can be mass movements of rock formations within landslide areas.

To provide safe operation, it is beneficial to equip pipeline structures with devices that can automatically control selected parameters important from their technical condition assessment. Having reliable data about the actual state of the structure, it is then possible to make optimal decisions [18] during the entire lifecycle regarding both engineering solutions and financial savings.

The most common approach described in the literature is to measure physical quantities along the pipeline, e.g., axial strains. This is the most sensitive solution for analysing deformations coming from external loads like movements of ground. The examples described hereafter form a brief stat-of-the-art-review, showing the applications based on the longitudinal measurements. However, it is worth underlining that strains coming from internal pressure in the pipeline are much larger than the longitudinal ones. That is why it is favorable and justified to develop a proven solution dedicated also for monitoring circumferential deformations. The research described in Section 4 is focused particularly on this aspect.

The most frequently measured physical parameters within composite pipelines are strains, displacements and temperatures. Knowing the values of these parameters and assuming an appropriate mathematical-physical model, it is possible to determine other values—such as axial forces or bending moments (in two planes). The calculated values should, however, be compared in real time to primally defined threshold values obtained from structural calculations. This approach allows for remedial actions to be taken immediately in the case of occurrence of potential threats. It is worth emphasising that the costs of preventing actions (e.g., costs of the monitoring, early warning system and resulting strengthening) are usually considerably lower than the costs of the failure consequences. In case of safety-critical structures, such consequences include not only structural costs, but mainly environmental and social costs [19]. What is more, there is also a risk of human fatalities, which absolutely makes the cost of prevention economically justifiable—human life is obviously priceless.

So far, the most commonly applied solution for the monitoring of pipeline structures is the installation of spot sensors (e.g., vibrating wire, Bragg grating) in selected cross-sections. Figure 1 presents selected results obtained from the automatic structural health monitoring system applied for pipelines in the Czech Republic [20]. Measured strain values were compared with threshold values. Figure 1a shows a situation wherein strains were dangerously approaching their limit, due to significant soil movements within the landslide area. Thus, it was decided to relieve the pipeline (excavating and backfilling with soil). Thanks to the immediate remedial action it was possible to ensure further safe operation. 

In contrast, Figure 1b presents the results of measurements performed within a pipeline section located in normal operating conditions. Here, only the effect of cyclical (annual) changes in temperature is clearly visible. Therefore, no intervention was necessary. However, it should be emphasised that in this case lack of action was based on conscious decision resulting from the objective measurement data. Therefore, the lack of action did not increase the failure risk and it was objectively justified. 

It is also worth noticing, that it is possible to convert the measured strains [με] into stress [MPa] values through appropriate physical model, which are more intuitive quantity for engineering interpretation.

In the case of spot measurements, three strain gauges are usually installed over the pipeline circumference every 120 degrees (Figure 2a). Such an approach allows for the analysis of effects related to both axial forces (Figure 2b) and two-plane bending (Figure 2c).

The application of conventional spot gauges for pipeline monitoring offers many obvious advantages, but its basic limitation is the relatively high costs and therefore the possibility of implementing measurements only in selected cross-sections (Figure 3). Such an approach does not allow for local damage detection (e.g., leakages, unsealings)—there is a high risk of overlooking threats that may occur between the measurement cross-sections.

The problem of gauge location can be entirely eliminated through application of distributed fibre optic sensing (DFOS) technique, as it enables measurement over the entire structural length (even over hundreds of kilometers). An example is the structural health monitoring of a pipeline transmitting saline in north-eastern Berlin. The research outcomes found in [21] present the example results of strain measurements gathered over a period of 2 years from the sensing cables arranged longitudinally on the pipeline every 120 degrees over a length of 500 m [21].

By means of the distributed fibre optic technique, it is possible to detect local damages [22], such as leaks or unsealing [23], that are usually related to changes in temperature. Another aspect is the mechanical loading caused by the ground movements. The typical location of fibre optic sensors on the pipeline is presented in Figure 4.

Although the longitudinal measurement approach is the most commonly applied in practical applications, it should be also noted that circumferential deformations could be much higher, especially when internal pressure in the pipeline is the main active load. The research described hereafter in Section 4 is focused on the analysis of circumferential strains and displacements (shape changes).

Another important aspect of distributed fibre optic sensing is the possibility to detect any deformations and local defects caused by the manufacturing process of fiber reinforced polymers. By using appropriate DFOS sensors, it is possible to measure strain and temperature of manufactured GRP components, what allows for evaluation of their quality already at the production stage [25,26].

## 3. Distributed Fibre Optic Sensing

### 3.1. Operation Rules

Distributed fibre optic sensing (DFOS) allows the measurements to be performed in a geometrically continuous way. Such measurements could be based on different physical phenomena (e.g., Rayleigh, Brillouin or Raman scattering) [27]. The specific technique should always be individually selected in reference to the requirements of a given project, as it influences the measured physical quantities (usually strain or temperature) and other measurement parameters like length of the individual gauge, spatial resolution, accuracy, time of measurements and many others.

In the work described hereafter, Rayleigh scattering was used due to its high spatial resolution and precision. The measurements of strain profiles over the entire length of optical sensors attached to the surfaces of composite collector were performed using optical backscatter reflectometer OBR4600 (from Luna Innovations). This device allows to obtain strain resolution of ±1 με. Lengths of the individual gauges over the length and their spacing (spatial resolution) were adopted at 10 mm (100 measurement points per 1 m of the sensor)—see configuration presented in Figure 5.

The distributed approach allowed for replacing thousands of traditional spot strain gauges arranged in series with a single optical fibre or sensor, and therefore for detection of local phenomena which are undetectable for conventional techniques [28,29].

### 3.2. Optical Fibres for Strain Measurements

Nowadays, optical fibres are usually made from highly purified silica glass with a specially designed structure that enables the transmission of electromagnetic waves. Optical fibres are produced from glass prisms which are heated and drawn into a thinner fibre. A standard telecom single-mode fibre just after drawing has a diameter of 125 μm, while the core is only 9 μm. For protecting the fragile glass against incidental mechanical damage and to enable appropriate management of the fibre, a primary coating is applied during the production process (usually with an external diameter of 250 μm).

The cross-section of standard optical fibre is presented in Figure 6a while Figure 6b shows the fibre on the spool delivered to the laboratory for the installation process. Depending on the final application of the optical fibre, different numbers and types of additional protective jackets are applied. For strain measurements purposes, all additional layers should always be deleted to not disturb the strain transfer mechanism. 

The laboratory research presented further in the article employed single-mode SM9/125 fibre only with its primary acrylate coating. All additional protective jackets were removed as they can disturb strain measurements as a result of the slippage phenomena between the intermediate layers. The efficient strain transfer mechanism is also a key aspect which should be strongly considered when designing DFOS sensors for civil engineering and geotechnical applications—see also Section 3.3. 

It is also worth underlying that in reference to composite structures, due to their production technology, it is possible to integrate optical fibres with composite members already at the stage of their prefabrication [31]. Such an approach has been successfully verified in laboratory and field research, including the aerospace industry [32] and civil engineering, e.g., within composite bridges [33] or composite deck panels for bridge engineering [34].

### 3.3. Fibre Optic Sensors for Geotechnics and Civil Engineering Applications

Distributed fibre optic sensing originated in the aerospace industry, but it is currently widely developed in other areas of engineering, including geotechnics and civil engineering. Optical fibres without protective jackets can be successfully applied in laboratory research, although in real operating conditions it is necessary to apply specially dedicated DFOS sensors. Standard sensing cables are made of a number of intermediate coatings, which does not allow for correct strain measurements. The slippage between particular layers results in inaccurate strain transfer from the monitored structure to the measuring core and finally the actual deformation state of the structure cannot be truly reflected. Thus, ongoing works are in progress to design and produce new and improved DFOS sensors for both strain and displacement measurements made especially for civil engineering and geotechnical applications [35].

In 2018, SHM System company [36] (Krakow, Poland) completed a research project called “Development of a new fibre optic sensor allowing for the determination of the vertical and horizontal displacements of the studied objects at the distances of up to 120 km”. This project was funded by the grant won at the National Centre for Research and Development. As a result, two types of fibre optic DFOS sensors were design and produced that can be successfully applied in structural monitoring of pipelines and the surrounding ground: EpsilonRebar and EpsilonSensor (Figure 7a) for axial strains measurements and 3DSensor (Figure 7b) for direct displacements measurements in 3D space. The application of these solutions in the frame of structural health monitoring of pipelines in longitudinal and circumferential direction are presented in Figure 8a,b, respectively.

## 4. Laboratory Research

### 4.1. Description of the Specimens

The research was performed at the laboratory of materials strength of the Department of Hydro-Engineering and Hydraulics at the Faculty of Building Services, Hydro and Environmental Engineering of the Warsaw University of Technology. Thanks to the application of helical filament winding technology, two specimens of GRP pipelines of circular cross-section were prefabricated. Such pipes are produced in a special form with circular or non-circular cross-sections through winding glass fibre mats and roving fabrics (sheets made of one or several two-directional layers of fibres) or spraying chopped fibres soaked in polyester or vinyl ester resin. Local strengthening of the cross-section (e.g., caused by ribbing) particularly occurs though application of one-way tapes, 3D fabrics, mats, and roving fabrics. The technology allows for obtaining any shapes of pipeline with variable wall thickness in the longitudinal direction and cross-section plane [37]. The primary advantage of such a process is the possibility to modify pipe parameters, adjusting them to the individual design requirements.

Two pipe specimens were tested during the studies described hereafter with the following geometrical parameters:Internal diameter—Ø700 mm; wall thickness—10 mm (Figure 9a);Internal diameter—Ø500 mm; wall thickness—15 mm (Figure 9b).

Optical fibres with their acrylate primary coating (Ø250 µm) were glued to the external and internal surface of the pipe using structural glue (two-component epoxy resin). One fibre route was designed in such a way to create three continuous loops over the full circumference with a spacing of 50/100/100/50 mm (Figure 10a). Each loop was designed and formed with overlapping sections (Figure 10b) to ensure strain measurement over the entire circumference without any interruptions. The surface of the specimens in the areas of fibre attachment was previously matted with sandpaper and carefully degreased.

Optical cables (pigtails) were then spliced on both ends of each fibre, allowing for their connection to the reflectometer through optical switch. The application of two active ends within the single fibre minimised the risk of data loss in case of fibre breakage. Finally, four pigtails were prepared for each pipe specimen, two for the internal and two for external surface.

### 4.2. Measurement Station and Course of the Study

Testing machine Heckert FPZ-100 with a regulated speed of one-directional piston displacement and maximum force equal to 100 kN (Figure 11a) was used to performed mechanical tests on the pipe specimens. Data from testing machine (including force and displacements) were registered automatically during the entire experment. What is more, reference data were also obtained directly from spot displacement sensor, which was installed directly below the internal pipe surface within its upper part (Figure 11b). The force sensor was mounted with a steel plate with dimensions of 30 × 35 × 8 mm to ensure even distribution of force over the upper part of the external surface of the specimen. An OSB plate was placed between the steel plate and the pipeline, with grooves enabling the safe operating of the fibres (measuring loops) under the area of direct force application (Figure 12a).

During research, the force was applied in a static way. After reaching the required force value through the press piston, the machine was stopped and then strain readings were taken by the OBR reflectometer. The optical switch with four active channels was used to accelerate the time of the single measurement session and thus the entire research. Reference readings were performed just before loading (with zero strain state). The measurements were done at 0.5 kN increments up to the force corresponding to the acceptable range of the pipe displacements (approx. 3 kN with 39.05 mm displacement for the Ø700 and 17.21 mm for the Ø500 specimen). The measurements were also performed during unloading up to the 0 force.

### 4.3. Example Measurement Results

#### 4.3.1. Strains Distributions

During data post-processing, the spatial resolution for DFOS results was assumed to be equal to 10 mm. This indicates 100 measurement points over 1 m of the fibre. Strain values can be displayed directly as a profile over the entire measuring section, including any from three loops. Some example data for both the internal (SI) and external (SE) measuring sections over the middle loop (2) are presented in Figure 13 in the length domain. Strain profiles are shown for selected load steps and data for the extreme force are shown in bold to facilitate their interpretation. The positive strain values correspond to the tension while negative ones to compression (values relative to the zero reading).

However, the above approach in data presentation is not intuitive because of the straight horizontal axis, which make the structural behaviour of circular cross-sections difficult to interpret. Hence, the same data are presented separately for external (Figure 14a) and internal (Figure 14b) surface using pie charts, corresponding to the real shape of the cross-section.

#### 4.3.2. Displacement Analysis

Based on the measured strain distributions in the subsequent load steps (at different force values), it is possible to determine displacements (changes in the collector’s shape relative to the original shape before loading). The primary method for this purpose is the application of analytical formula, which allow for curvature assessment. The basic equation in cartesian (*x*, *y*) coordinate system notation, coming from differential geometry and describing the curvature of any plane curve, is as follows:(1)$$\kappa =\frac{1}{R}=\frac{\frac{d^{2}y}{dx^{2}}}{{\left(1+{\left(\frac{dy}{dx}\right)}^{2}\right)}^{\frac{3}{2}}}$$
where: 

κ—curvature;R—radius of curvature;y—displacement.

Although the analytical solution does not require any additional material assumptions, loadings or other boundary conditions, it has no sufficient accuracy in real applications due to the local material imperfections and discontinuities, as well as limited accuracy of strain measurements (resulting from both the sensor and data logger).

Another practical method for estimating displacements (changes in shape) is by numerical simulation by finite element model (FEM)—Figure 15. Nowadays, it is a basic design tool so it is favorable to use it also during measurement data interpretation. Strain results can be compared with simulations in a way allowing for adjusting the model parameters (e.g., stiffens or boundary conditions) to obtain better compliance. Such a model, with increased reliability, could be further used to estimate other parameters, e.g., displacements with better accuracy than before calibration. Example results from FE analysis are summarized in Table 1.

However, it should be remembered that due to the assumptions and simplifications concerning both the geometry of the structure, material properties, boundary conditions and cooperation with the surrounding ground, this approach, especially for real in situ applications, would be still uncertain [38] and very often not sufficient.

Numerical analysis requires knowledge about material properties, loadings, geometry and boundary conditions. Another approach is based on direct displacement calculation based on measured strain values. There are a few methods described in the literature [38], but one of the most intuitive and with practical potential is the trapezoidal one. What is extremely important, there is no need to know material parameters (e.g., elastic modulus) of the structure. The input data are only strain profiles over the entire length measured by two optical fibres in known spacing and boundary conditions (e.g., displacement and rotation only in one point, for instance at the base). In case of the pipeline deformation analysis, these two fibres are located over internal and external surface, while the thickness of the pipe is known. Calculations also depend on spatial resolution, which is a post-processing parameter. Detailed description of the presented approach is described in [30] and the general equation can be written as follows: (2)$$u_{v}=f\left(\varepsilon_{I},\varepsilon_{E},t,b,bc\right)$$
where:*u_v_*—displacement profile (mm) over measuring length;*ε_I_*—strain profile [με] over internal measuring length;*ε_E_*—strain profile [με] over external measuring length;*t*—wall thickness [mm] (distance between optical fibres);*b*—spatial resolution [mm] (base length of individual sensors over length);*bc*—boundary conditions (e.g., initial displacement and rotation angle or displacements in two known locations).

Trapezoidal method is the basic principle of operation of distributed fibre optic 3DSensor (see also Figure 8b [24]), where four optical fibres are arranged at exactly known positions around the sensor’s composite core. The calculation method applied during research is described in a number of papers [30,33,34]. It is based on the analysis of deformations of individual trapezoid which is defined by spatial resolution and the distance between internal and external fibre. After calculation of displacement within the individual trapezoid, it goes to the next one, adding up the displacement values. Originally this approach was used specifically for applications where initial configuration was approximately a straight line (e.g., along bridge spans, roads, embankments, dams, pipelines, landslide areas, slurry walls, etc.) However, a more challenging situation appears when analysing pipeline deformation within their cross-section plane. The initial configuration is approximately circular and thus spatial resolution of measurements within the internal and external surface should be adopted individually to obtain agreement between the number of measuring points within both surfaces. This is due to different base lengths of individual gauges depending on the location (internal and external). This geometrical effect is presented schematically in Figure 16. Compensating of the differences in internal and external surface spatial resolutions is done during data postprocessing (by assuming appropriate resolution in data logger software or by formulating an appropriate approximation function).

Estimation of displacement based on strain measurements and trapezoidal method gave very good compliance with comparison to the reference spot displacement sensor (the mean standard deviation of error was less than <0.4 mm while mean error less than 0.1 mm). Moreover, calibrated FE model indicated very good agreement with DFOS results (the mean standard deviation of error was less than <0.4 mm while mean error less than 0.2 mm). A detailed data summary is presented in Table 1. 

The algorithm was checked only in the one reference point. However, it aggregates the displacement values from the base point over entire circumference, so all the points over length influence the final compliance. Thanks to this approach, DFOS data processing offers a possibility for reconstructing the structure’s actual shape (deformation along the entire circumference), while the spot gauges are able to provide information only in one selected point. 

Example displacement calculation results aimed at determining the actual shape of the collector at subsequent load steps during research are shown in Figure 17. Displacements were estimated directly based on internal and external strain data presented in Figure 13 and Figure 14. Herein, data are presented in a pie chart for the most intuitive engineering interpretation. The actual shape of the collector is represented by the horizontal and vertical coordinates in a Cartesian system. However, the displacement values are 5 times scaled to make the plot clear and readable.

## 5. Conclusions

The advantages of distributed fibre optic sensing (DFOS), not achievable for conventional spot techniques, allow for obtaining comprehensive and reliable information for actual assessment of the technical condition of different types of composite structures. Nowadays, this measurement approach is used in the area of aerospace engineering [39], but more and more often also in civil engineering [40,41] and geotechnical applications. DFOS could be successfully applied within long-distance structures such as pipelines and collectors over distances of several hundred kilometers. However, it may also be useful for precise short-distance applications like the one described in the article—related to the circumferential deformations of the pipeline cross section. It is worth noticing that the circumferential strains caused by internal pressure in the pipeline could be much higher than the longitudinal ones.

The article presents laboratory research where the shape of the circular collector specimen was successfully estimated based on strain measurements during load tests. Based on the state-of-the-art review as well on obtained measurement results, the following insights can be drawn:(1)The novelty of presented measurement approach lies in the possibility of parallel calculations of displacements (changes in shape) with satisfying accuracy directly based on measured strain profiles.(2)To ensure appropriate displacement accuracy, the key aspects are both precise physical arrangement of the sensors within the structure, as well as the appropriate calculation approach (trapezoidal method based on at least two strain profiles [33,34,38]).(3)Data were compared with independent reference technique and numerical simulations, obtaining very good compliance. Mean error level was less than 0.2 mm. Thus, the efficiency of the proposed method for DFOS data post-processing was confirmed.(4)Application of distributed fibre optic sensors allows to obtain comprehensive knowledge about the deformation state of the collectors and pipelines. This knowledge includes not only structural strains, but also displacements and could be successfully used for assessment of structural condition.(5)Because of the extremely low diameter of optical fibres only in their primary coatings, compared to the size of composite structural members, such fibres can be integrated inside the composite laminates during their production process [42]. This will allow for creating smart structures able for auto-diagnosis [34].(6)Due to the very low costs of fibre optic sensors and their advantages over conventional spot techniques, it is recommended to equip safety-critical pipelines with this type of measurement solution. Measurements could be performed during construction, repair works and further operation of the collectors modernised with GFRP panels.

Structural health monitoring based on distributed fibre optic sensing can significantly increase the structural safety and improve the maintenance procedures by local damage detection. In many cases, it is not advisable to design and build very expensive conservative structures. It is more economical and technically justified to monitor the strain-stress state and react immediately in the case of a dangerous, but unlikely events like leakages. Works on smart composite members equipped with DFOS sensors are still in progress in many engineering and scientific institutions worldwide. The research presented in this article is part of these global trends.

## 6. Patents

US patent, Application number: 15/849,804, Patent number: US 10,620,018 B2, Title: Method for measuring the displacement profile of buildings and sensor therefor, Application date: 21 December 2017, Publication date: 14 April·2020, Applicant: SHM System Sp. z o.o., Sp. komandytowa, Inventors: Bednarski Ł., Sieńko R.

Polish patent (PL), Application number: P.412838, Patent number: Pat.235392, Title: Method for continuous measurement of the building objects relocation profile and a sensor for execution of this method, Application date: 24 June 2015, Publication date: 13 February 2020, Applicant: SHM System Sp. z o.o., Sp. komandytowa, Inventors: Bednarski Ł., Sieńko R.

EP3314202A1, Title: Method for measuring the displacement profile of buildings and sensor therefor, Application number: EP16744907A, Application date: 17 June 2016, Publication number: EP3314202A1, Publication date: 2 May 2018, Applicant: SHM System Sp. z o.o., Sp. komandytowa, Inventors: Bednarski Ł., Sieńko R.

Canadian Intellectual Property Office, Patent Application: CA 2989301, Title: Method for measuring the displacement profile of buildings and sensor therefor, PCT Filling date: 17 June 2016, Open to public inspection: 29 December 2016, Applicant: SHM System Sp. z o.o., Sp. komandytowa, Inventors: Bednarski Ł., Sieńko R.

International Publication Number: WO 2016/209099 A1, Title: Method for measuring the displacement profile of buildings and sensor therefor, Publication date: 29 December 2016, Applicant: SHM System Sp. z o.o., Sp. komandytowa, Inventors: Bednarski Ł., Sieńko R.

## Figures and Tables

**Figure 1 sensors-21-05904-f001:**
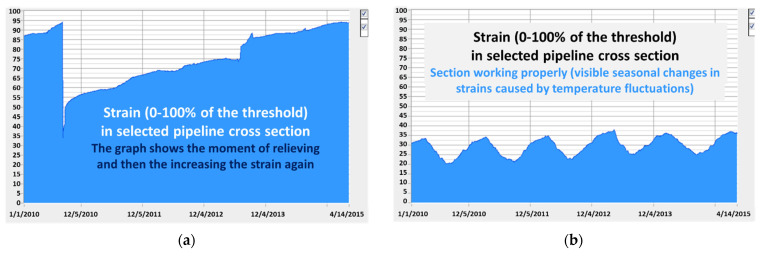
Exemplary plots from the pipeline structural health monitoring system in the Czech Republic showing strain development over time in relation to the threshold value (0–100%) for: (**a**) the section operating in the landslide area; (**b**) the section operating under normal conditions exposed only to thermal effects [20].

**Figure 2 sensors-21-05904-f002:**
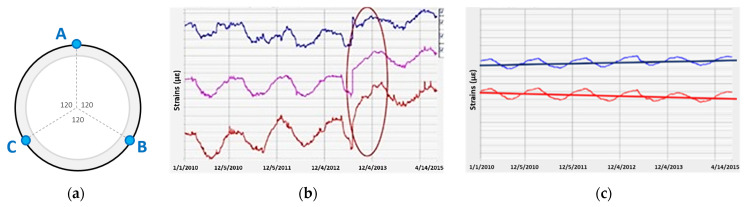
(**a**) Typical arrangement of sensors in the pipeline cross-section; (**b**) example of measurement results indicating the sudden increase in axial force [20]; (**c**) example of measurement results indicating a gradual increase in bending (stress increase both on the tension and compression side) [20].

**Figure 3 sensors-21-05904-f003:**
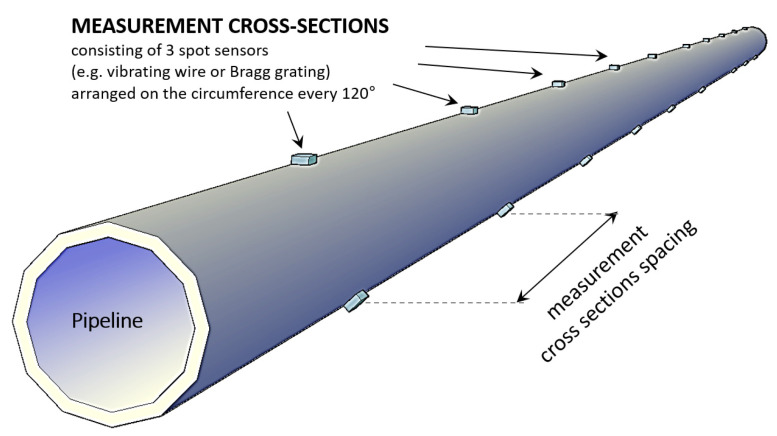
Typical concept of spot monitoring of the pipeline.

**Figure 4 sensors-21-05904-f004:**
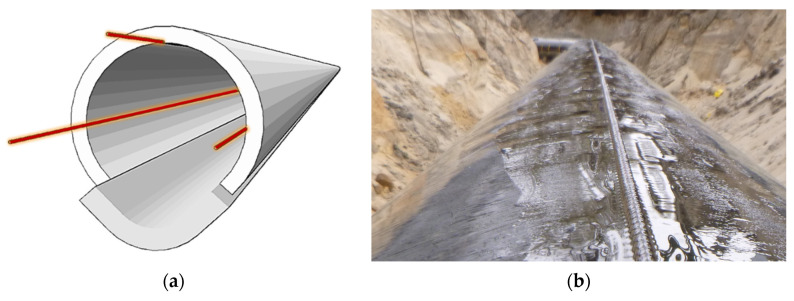
(**a**) Typical location of optical fibre sensors within the pipeline (usually three sensors for 3D space calculations); (**b**) example application of DFOS strain sensor in the upper part of the gas pipeline [24].

**Figure 5 sensors-21-05904-f005:**
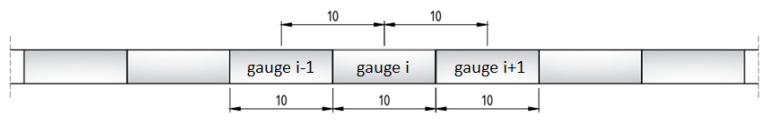
Parameters of distributed measurements applied during laboratory tests [mm] [30].

**Figure 6 sensors-21-05904-f006:**
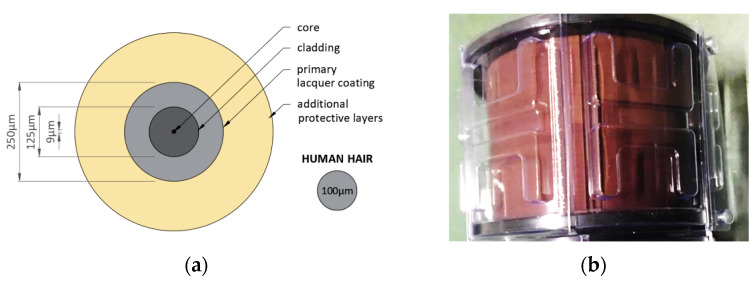
Standard single-mode optical fibre: (**a**) cross-section, (**b**) the view of fibre on a spool, which was delivered to the laboratory for installation.

**Figure 7 sensors-21-05904-f007:**
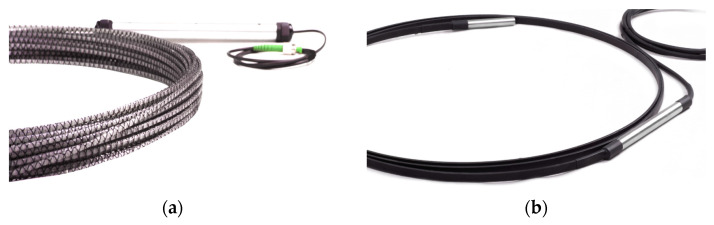
Examples of monolithic DFOS sensors for civil engineering and geotechnical applications: (**a**) EpsilonSensor for axial strains and cracks measurements; (**b**) laboratory version of 3DSensor for 3D displacement measurements [24].

**Figure 8 sensors-21-05904-f008:**
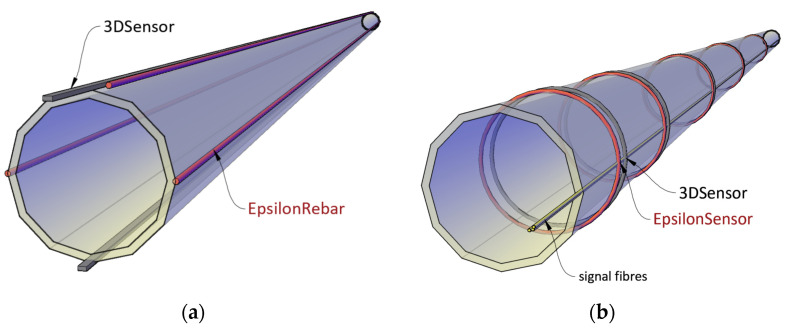
Application of composite and monolithic DFOS sensors for strain (EpsilonRebar and EpsilonSensor) and 3D displacement (3DSensor) measurements over: (**a**) longitudinal and (**b**) circumferential direction.

**Figure 9 sensors-21-05904-f009:**
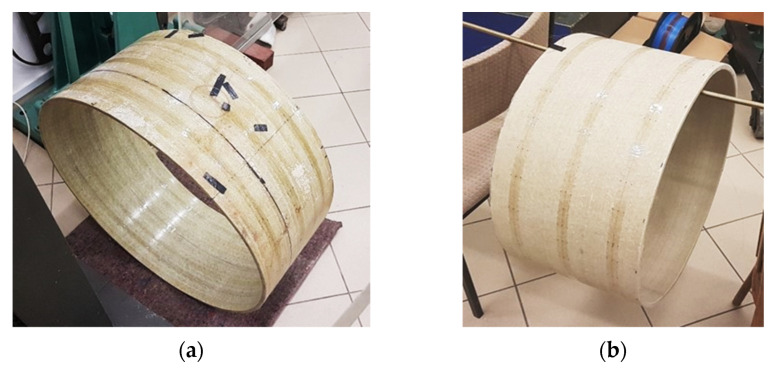
GRP pipe specimens prepared for testing: (**a**) DN 700 mm; (**b**) DN 500 mm.

**Figure 10 sensors-21-05904-f010:**
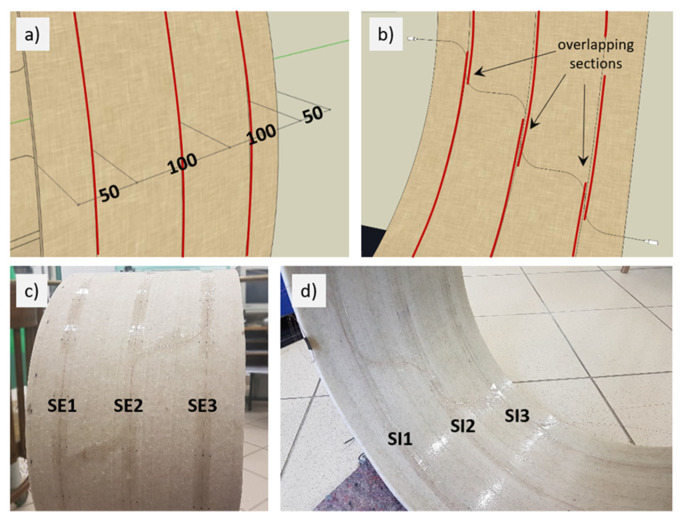
Optical fibre installation: (**a**) arrangement of optical fibre loops on the external surface of the pipe; (**b**) view of overlapping sections; view of fibres on the external (**c**) and internal (**d**) surface of the pipe with assumed names.

**Figure 11 sensors-21-05904-f011:**
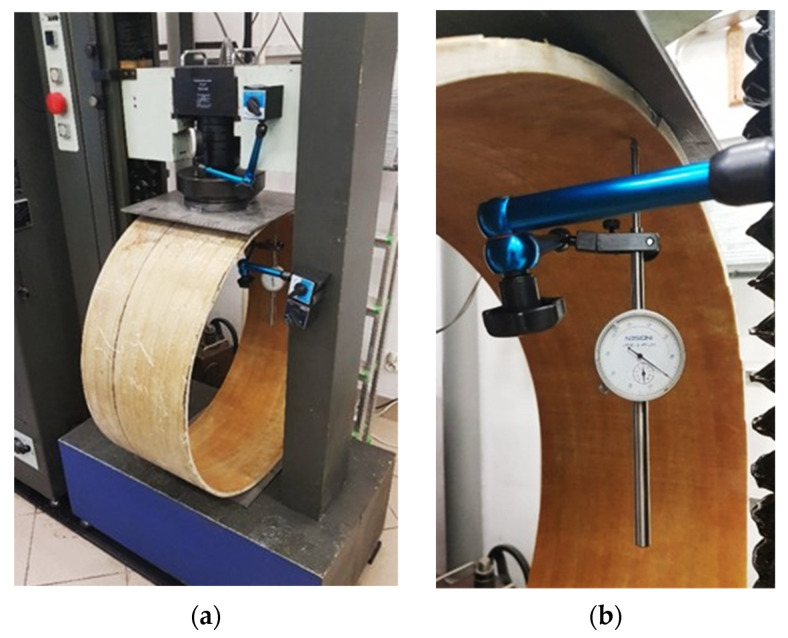
Measurement stand. (**a**) General view; (**b**) close-up of the reference displacement sensor.

**Figure 12 sensors-21-05904-f012:**
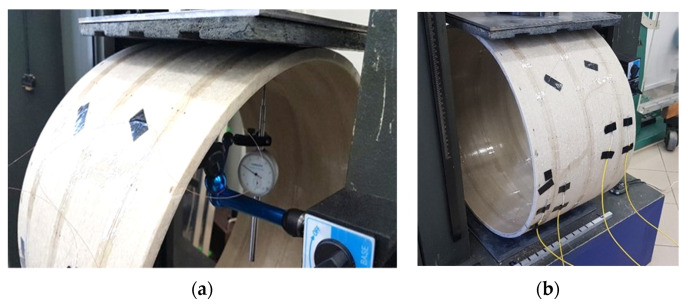
(**a**) The view of OSB board with grooves enabling the safe conduct of the fibres under force area; (**b**) measurement station with a specimen placed in the testing machine and optical fibres connected to the reflectometer.

**Figure 13 sensors-21-05904-f013:**
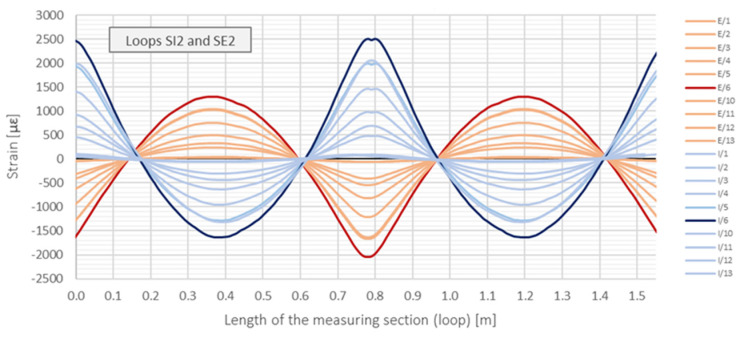
Example strain distributions [με] over the internal and external circumference of the GFRP Ø500 pipe in selected load steps for the loops SI2 and SE2 (straight line plot).

**Figure 14 sensors-21-05904-f014:**
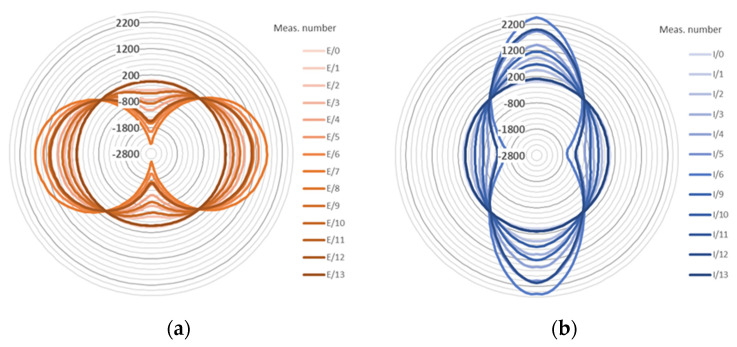
Example strain distributions [με] over the external (**a**) and internal (**b**) circumference of the GRP Ø500 pipe in selected load steps for the loops SI2 and SE2 (pie chart).

**Figure 15 sensors-21-05904-f015:**
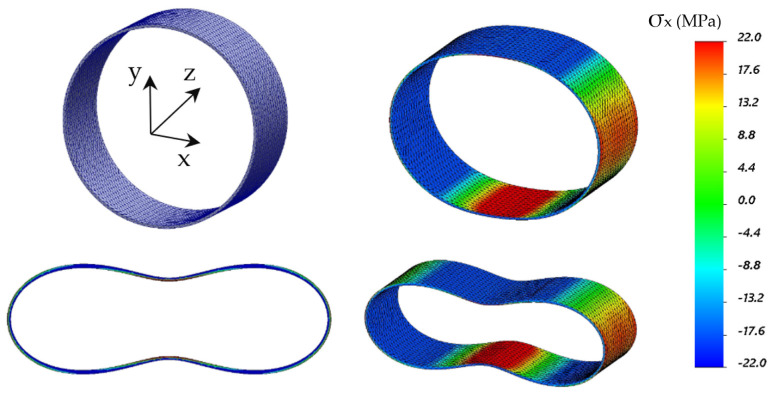
Spatial visualization of the GRP pipe deformation based on FEM analysis caused by the external loading of 3 kN (color map corresponds to strains: blue—compression, red—tension; material properties: elastic modulus E_x_ = E_y_ = 8.0 GPa, E_z_ = 7.8 GPa, Poisson’s ratio ν = 0.2).

**Figure 16 sensors-21-05904-f016:**
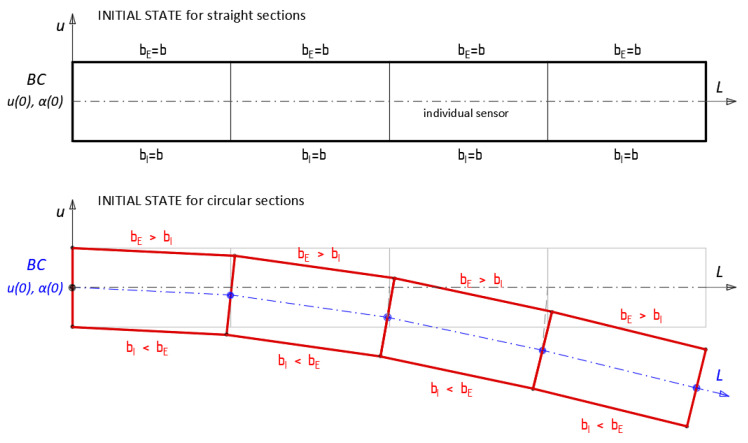
Graphical interpretation of trapezoidal method used. First assumptions for displacement algorithm depending on the initial configuration (straight line or circular cross-section).

**Figure 17 sensors-21-05904-f017:**
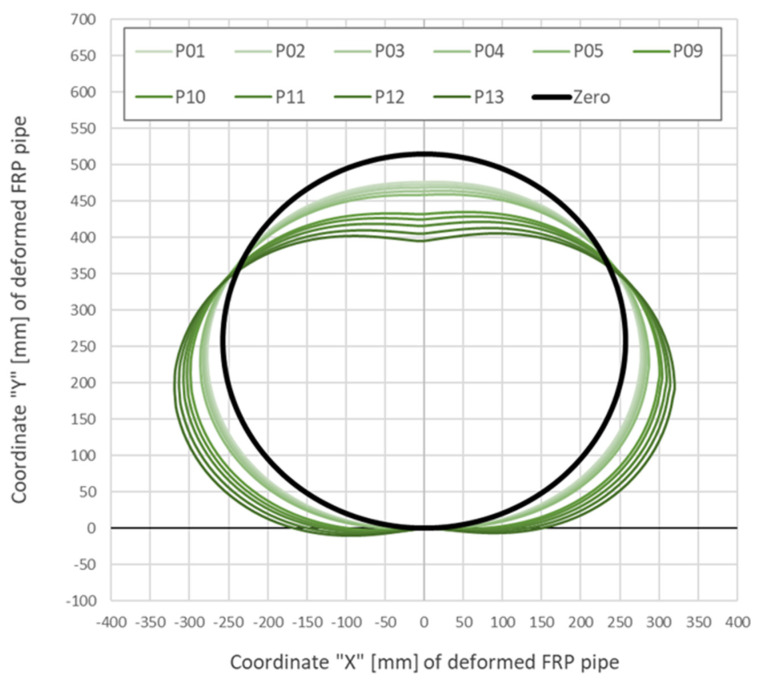
Calculated displacements [mm] (shape changes, coordinates) of the GFRP Ø500 specimen during selected load steps (scaled).

**Table 1 sensors-21-05904-t001:** Summary of displacement calculations in comparison to reference spot gauge.

No. of Load Step	0	1	2	3	4	5	6
Force [kN]	0.0	0.5	1.0	1.5	2.0	2.5	3.0
DFOS displ. [mm]	0.00	2.17	5.06	8.31	10.67	15.22	17.65
Ref. displ. [mm]	0.00	2.70	5.44	8.17	11.06	14.95	17.21
FEM displ. [mm]	0.00	2.21	5.02	7.99	10.90	14.30	17.70
Diff. (REF—DFOS) [mm]	0.00	0.38	−0.14	0.39	−0.27	−0.44	0.38
Diff. (FEM—DFOS) [mm]	0.00	−0.04	−0.32	0.23	−0.92	0.05	−0.04
Mean error (REF—FDOS) [mm]	0.08						
Stdv. (REF—DFOS) [mm]	0.37						
Mean error (FEM—FDOS) [mm]	−0.16						
Stdv. (FEM—DFOS) [mm]	0.38						

## Data Availability

The data presented in this study are available on request from the corresponding author.

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
