# Peer review of "Distributed Fibre Optic Sensing (DFOS) for Deformation Assessment of Composite Collectors and Pipelines"

_sensors, 2021, doi:10.3390/s21175904_

Round 1
Reviewer 1 Report
The submitted manuscript deals with the application of optic fiber technology to implement a distributed sensing system to monitor the deformation of structure in fiber reinforced polymers. The authors provided a wide synopsis of the state-of-the-art about optic fiber based monitoring systems, described the experimental activity performed on cylindrical samples produced by filament winding, and commented on the achieved results. Moreover, a numerical model has been implemented to evaluate the local displacements taking as an input the strain field.
The manuscript is well written, and in the referee’s opinion, the topic is appealing for the readership of Sensors and for researchers involved in fiber reinforced polymers manufacturing/designing/monitoring.
Some minor remarks are following listed:
- Sensing of composite materials assumes particular relevance whereas fiber reinforced polymers are applied to structural building elements. Useful synopsis of the main applications of fiber reinforced composites can be found in “Pultruded materials and structures: A review”, Journal of Composite Materials 2020, by Vedernikov et al.
- On the proof analyzed in the present peer review, Figures 1 and 2 are not completely readable due to the low quality.
- Please, double-check the link provided in reference [19]. I cannot find there the images reported in Figures 1 and 2. Add the access date to the reference.
- In the referee’s opinion, the adoption of optical fiber sensors for the detection of manufacturing induced deformation and defects during the production of fiber reinforced polymers is another relevant aspect. This has been recently documented in the following papers:
- Oromiehie et al., Characterization of Process-induced Defects in Automated Fiber Placement Manufacturing of Composites Using Fiber Bragg Grating Sensors, 2017
- Tucci et al., Strain and temperature measurement in pultrusion processes by fiber Bragg grating sensors, 2018
- Figure 5 was already reported in a previous manuscript (Bednarski et al., New Distributed Fibre Optic 3DSensor with Thermal Self-Compensation System: Design, Research and Field Proof Application inside Geotechnical Structure, Sensors 2021). Please, add a reference to this work in the caption of Figure 5.
- The authors are invited to clarify the method adopted to interrogate the circumferential optical fibers (Figure 8b) during the in-service life, as they do not present a free end to provide stimuli or detect feedback signals.
- In section 4.1 Description of the specimens, the authors are encouraged to describe in detail the reinforcement architecture, fiber volume fraction, and materials adopted in the production of the samples.
- Please, provide add colorbar-scales to promote the readability of Figure 16.
Author Response
Dear Reviewer,
thank you for all the comments and suggestions.
Please find attached the answers to your review.
Best Regards,
Bartosz Bednarz

Reviewer 2 Report
The article presents a laboratory and numerical simulation of the potential of DFOS for strain and deformation measurements of cylindrical pipelines and collectors. The work has merits as a scientific contribution. However, the presentation is rather sub-standard. The reviewer has given some comments below, which should be considered for updating the whole document for publication in the future.
- The reviewer believe title would be more suitable in the form, “Distributed fibre optic sensing (DFOS) for deformation assessment of composite collectors and pipelines”
- Line 16 – Should be “laboratory tests, where DFOS”
- Line 17 – “ex-tremely” should be corrected.
- Line 20 – “collec-tors.” Should be corrected.
- Line 22 – “ad-just” Should be corrected. This applied to rest of the document.
- The phrase in line 20-21, “It should be also noted that even advanced numerical simulations are not able to truly reflect real ground conditions and all construction stages.” would be more appropriate for the abstract in the form, “It is challenging to truly simulate real ground conditions at all construction stages”.
- In general, the later part of the abstract seems general. It should reflect the results and outcomes of this study.
- Sentences in Line 119 – 121 should be replaced with more meaningful sentence.
- The reviewer believes, only one figure among Figure 1 and Figure 2 is necessary to explain the significance since they indicate a trend in deviation in actual data compared to predicted or expected data. Mention in the text is thought enough. Same opinion on Figure 4.
- The Introduction should be revisited for coherence in writing and storytelling.
- Section 3 is written in a ‘novice’ way. The author (s) should keep their storytelling streamlined. Please read some reputed journals to observe how the sections are written. Section 3 is a combination of describing what the technology you have chosen, why and how.
- Section 4.3 heading “Exemple”?
- What simulation software package is used? Details?
- No material or GFRP properties is given.
- The section 4.3 indicates the merit of this work, which is worthy of a publication. The results are presented in a disorganized way.
- No real tangible conclusion is presented except long-winding two paragraphs just explaining the background and other references. The conclusion should indicate results and comparison in number or at least in clear textual distinctions.
Author Response
Dear Reviewer,
thank you for all the comments and suggestions.
Please find attached the answers to your review.
The manuscript with the changes applied has been re-uploaded.
Best Regards,
Bartosz Bednarz

Reviewer 3 Report
This article presents a new approach to control and monitor of Glass Fiber Reinforced Polymer (GFRP) collectors and pipelines, using distributed fiber optic sensors DFOS. Through experimental tests to simulate the deformation of the pipeline, the strain at each point can be measured. Analyzing the measured strain distribution, the approximate shape of the pipe in its cross-section can be calculated.
This article innovatively studies the circumferential strain of the pipe and the displacement can be calculated directly based on the measured strain distribution. The research method is innovative and has reference application value.
However, the following issues need to be modified:
- The error analysis only uses the displacement of one point as a comparison. It is recommended that the actual pipe shape and the calculated pipe shape be drawn in the same diagram for comparison.
- The method of applying force to the simulated pipe is only the method of applying force vertically. It is recommended to apply force from multiple angles to verify the accuracy of pipeline displacement calculation.
- Regarding numerical simulation, only the advantages and disadvantages are analyzed. It is recommended to do several numerical simulation experiments to facilitate comparison with laboratory experiments.
- The text label on Figure 2 is too small to be clear.
Author Response

(The authors gave the same response as above.)

Round 2
Reviewer 2 Report
The authors have revisited most of the issues raised. The manuscript is more polished now. I still see some scope of improvements. However, I would request to revisit the abstract and conclusions again. The first conclusion (1) does not make any sense. Everyone knows that DFOS can measure strain, and temperatures. This is not a conclusion! I would request the first author to consult other co-authors to revisit the conclusion.
I have re-written the abstract based on what I understood from the existing abstract. You can update or edit it as necessary. I request to do the same for conclusions; please see an existing top tier journal. Hope this helps:
" Due to the low costs of distributed optical fibre sensors (DFOS) and the possibility of their direct integration within layered composites members, DFOS technology has considerable potential in structural health monitoring of linear underground infrastructures. Often, it is challenging to truly simulate the actual ground conditions at all construction stages. Thus, reliable measurements are required to adjust the model and verify theoretical calculations. The article presents a new approach to monitor displacements and strains in Glass Fiber Reinforced Polymer (GFRP) collectors and pipelines using DFOS. The research verifies the effectiveness of the proposed monitoring solution for health monitoring of composite pipelines. Optical fibres were installed over the circumference of composite tubular pipe, both on the internal and external surfaces while loaded internally. Analysis of strain profiles allowed for calculating the actual displacements (shape) of the pipe within its cross-section plane using the Trapezoidal method. The accuracy of proposed approach was positively verified both with reference spot displacement transducer as well as numerical simulations using finite element method (FEM). DFOS could obtain comprehensive view of structural deformations, including both strains and displacements under internally applied load. The knowledge gained during research will be ultimately used for renovating existing collectors."
Author Response
Dear Reviewer,
We sincerely thank you for the provided comments on the manuscript.
In fact, conclusion 1 was a truism, so we decided to remove it according to your suggestion.
In the submitted abstract proposal, we have only changed the method of force application. The force was applied externally. The rest of the abstract fully describes the essence of our manuscript. Thank you very much.
The resubmitted manuscript includes the comments you have provided.
Kind regards,
Bartosz Bednarz
Reviewer 3 Report
The manuscript has been largely improved. I accept the publication of this manuscript without further review
Author Response
Dear Reviewer,
We sincerely thank you for recognizing our manuscript as a scientific contribution.
Kind regards,
Bartosz Bednarz
This manuscript is a resubmission of an earlier submission. The following is a list of the peer review reports and author responses from that submission.
Round 1
Reviewer 1 Report
The paper is structured in a very confuse way. Up to section 4, the authors deal with the monitoring of pipelines with DFOS in the longitudinal direction. The introduction and state of the art are dealign with this. However, in section 4, authors start talking about monitoring in the cross-section of the pipeline, which is not related to the previous sections.
The paper does not explain clearly how the displacements are calculated from the strain profiles. They just mention that the calibration of a FEM is carried out in the way that the calculated displacement is close to the experimental one. However, experimental displacement is only measured at 1 single point and therefore, many calibration results are possible to fit this value. The text in lines 336 to 346 is not clarifying the procedure at all. This needs further explanations.
The authors should be aware that in a circular shape, the calculation of the displacement at any point can be easily solved by the integration of the differential equations and the solution can be found in text books of mechanics of continuous media, not being necessary a numerical model. The analytical solution does not need any assumption regarding the load, the boundary conditions, the stiffness of the material and so on. The only input needed is the curvature at any point. As in the tests, the strain is known at the exterior and interior surfaces of the pipe, the curvature at the points of measuring (at 5 mm interval) can be easily calculated and from this, the displacement at the measured points without any assumption about boundary conditions, material properties,...., because all this is already implicitely considered in the measured value of strain.
In summary, the paper is not providing any advance in the state of the knowledge, is very confusing and does not consider relevant methods to calculate the displacements. The only relevant data is the use of DFOS for measuring strain in GFRP material, something that is already well known. Because of that, my decision is to reject the paper.
Other comments:
1.- In line 88: "It is worth emphasising that the costs of preventing actions (costs of the monitoring and early warning system, costs of strengthening) are always considerably lower than the costs of the failure consequences" The authors should provide evidences and references to support this strong statement
2.-Reference 12 is incorrect
Author Response
Dear Reviewer,
thank you for all the comments and constructive suggestions.
Please see the attachment, where you can find our answers.
Best regards,
Bartosz Bednarz

Reviewer 2 Report
The article describes an application of high resolution strain monitoring to pipeline sensing. The article is well written, but improvements shall be made in several areas:
- The article gives the wrong impression that the laboratory results can also be achieved for long pipelines. This is not true as high resolution sensing is only possible for short ranges. This limitation is neglected in the article but has to be mentioned
- Lines 137 ff.: Regarding leakages, the sensing cable is very often not directly attached to the pipeline but embedded into the ground. This installation method is on one hand cheaper and on the other hand, mounting the cable directly on the bottom or top of the pipeline is not the best spot for leakage detection. If the leak occurs e.g. at 3 or 9 o’clock the liquid may pass by the fibre. Therefore the cable is usually placed at a certain distance from the pipeline. As a consequence Figure, does not show the best locations for leakage detection.
- Regarding the test setup: Why was only one reference sensor used and not several sensors distributed around the circumference?
- References: Section 4.3.2. does not contain any references on how the shape is calculated from the strain readings. Either some references are not at the right location in the paper or references are missing
- Lines 353 What was the size of the displacement and what is the error in percentage of the applied deformation?
- The most interesting aspect as how the shape can be calculated from when the initial status is already curved is not explained. The explanation in lines 340-345 is rather vague. Where the measurements only used to calibrated the finite element model. This this approach only work when the direction of the applied load is known? What about situations where there is bulging of the pipe? Can this also be solved => This section has to be extended significantly
Author Response

(The authors gave the same response as above.)

Round 2
Reviewer 1 Report
In their response, the authors claim: “The aim of the article is not to describe the mathematical algorithm in detail, but thanks to this new references, the particularly interested readers will be able to find the most relevant publications from 2021. Application of DFOS strain and displacement measurements within a proven, real case study, is truly novel approach…….”
After looking to the new references added where the so-called trapezoidal method applied by the authors as the mathematical algorithm, this reviewer has realized that what they are doing is just integrating the curvatures by a numerical approach based on the trapezoid rule. Because the curvature values are obtained in specific points, separated by the spatial resolution of the fibers, the function to be integrated is given by points and therefore, the approach to the real value of the integral is to apply a numerical integration (what they call the trapezoidal method). This in fact is not new as the theoretical basis on how to obtain the displacements from the curvatures (obtained from 2 strain profiles at different levels) was already formulated by Navier and Bresse in the 19th century. Therefore, this is not a novel contribution, as both the Navier-Bresse equations as well as the methods of numerical integration are very well known from many years ago.
Based on their response, the main conclusion and novelty of the paper is, as the authors write in the conclusions chapter: “However, the novelty of the measurement solution described in the article lies in the possibility of calculation displacements directly based on measured strain profiles. Thanks to the precise arrangement of sensors within the structure, as well as the appropriate calculation approach (trapezoidal method [30,31, 35]), it is possible to determine actual displacement profile both over the entire collector length and within the individual cross-sections”.
This is not really new, not only according to my previous comment above, but also because this method (calculation of displacements based on 2 strain profiles to obtain the curvature) has been already applied successfully by other researchers working on the subject of DOFS ( see for instance the papers published by Neil Hoult and his research group). Therefore, all these previous works had to be identified in the literature search and mentioned in the state of the art.
The authors also comment in their response: “However, according to the authors’ experience, analytical approach for calculating displacements is not accurate enough when working on real measurements data. This is due to the local material imperfections and discontinuities, as well as limited accuracy of strain measurements (resulted from both the sensor and data logger)”
I really can not accept this statement because what the authors are actually doing in the paper is applying the analytical approach (solved with a numerical integration because the integrand is only known at certain points) to calculate the displacements using real strain data.
Now the paper is not so confusing as it is clearly explained in the introduction that although DOFS have been used in the monitoring of pipelines in longitudinal direction, the paper is interested in the cross-sectional behavior.
Therefore, my decision is to reject the paper because as stated in my previous review: the paper is not providing any advance in the state of the knowledge, and the only relevant data is the use of DOFS for measuring strain in GFRP material, something that is already well known.